# Diagnostic accuracy of perinatal post-mortem ultrasound (PMUS): a systematic review

Susan Shelmerdine ,[1,2] Dean Langan,[2] Neil J Sebire,[2,3] Owen Arthurs[1,2]

► Additional material is published online only. To view please visit the journal online (http://dx.doi.org/10.1136/bmjpo-2019-000566).

[1]Department of Clinical Radiology, Great Ormond Street Hospital for Children, London, UK
[2]UCL GOSH ICH, London, UK
[3]Department of Paediatric Pathology, Great Ormond Street Hospital for Children, London, UK

**Correspondence to**
Dr Susan Shelmerdine; susan. shelmerdine@gosh.nhs.uk

## ABSTRACT

**Objective** Ultrasound is ubiquitous in live paediatric imaging; however, its usage in post-mortem setting is less established. This systematic review aims to evaluate the diagnostic accuracy of paediatric post-mortem ultrasound (PMUS).

**Design** MEDLINE, Embase and Cochrane Library databases were queried for studies published between 1998 and 2018 assessing PMUS diagnostic accuracy rates in children<18 years old, using autopsy as reference standard. Risk of bias was assessed using Quality Assessment of Diagnostic Accuracy Studies 2. A bivariate random-effects model was used to obtain combined mean estimates of sensitivity and specificity for different body systems.

**Results** Four studies were included, all relating to ultrasound for perinatal deaths. The mean diagnostic sensitivity and specificity for neurological abnormalities were 84.3% (95% CI: 70.8% to 92.2%) and 96.7% (95% CI: 86.5% to 99.3%); for cardiothoracic abnormalities 52.1% (95% CI: 27.6% to 75.5%,) and 96.6% (95% CI: 86.8% to 99.2%); and for abdominal abnormalities 78.4% (95% CI: 61.0% to 89.4%) and 97.3% (95% CI: 88.9% to 99.4%). Combining all body systems, the mean sensitivity and specificity were 73.3% (95% CI: 59.9% to 83.5%) and 96.6% (95% CI: 92.6% to 98.4%).

**Conclusions** PMUS demonstrates a reasonable diagnostic accuracy, particularly for abdominal and neurological abnormalities, although cardiac anomalies were less readily identified.

**Trial registration number** CRD42018106968.

### What is known about the subject?

► Whole-body post-mortem ultrasound (PMUS) is being performed in several European centres for perinatal deaths.
► Perinatal PMUS can be a useful alternative technique for non-invasive autopsy, where MRI is unavailable.

### What this study adds?

► Perinatal PMUS provides a high diagnostic accuracy for neurological and abdominal pathologies.
► Assessment of perinatal post-mortem cardiothoracic anomalies is challenging and less accurate.

## INTRODUCTION

Post-mortem imaging techniques in children have gained popularity over the past decade, particularly given the decline in parental consent rates for invasive autopsy.[1 2] Factors for this are multifaceted but include religious, emotional and communication barriers combined with the continued desire for further information to aid in future pregnancy management and bereavement counselling.[3]

The majority of studies relating to post-mortem imaging in children have related to post-mortem MRI (PMMR), with a recently published systematic review[4] reporting sensitivity rates of 0.73 (95% CI: 0.56 to 0.84) in a pooled sample of 953 children across eight studies. For post-mortem CT imaging, there have been fewer studies in children compared with adults, with the majority reporting CT to be of lower yield in identification for the cause of death in non-suspicious (ie, 'natural' deaths) and perinatal losses,[5–7] with a slight increase in benefit within a forensic setting.[8 9]

Ultrasound, while ubiquitous in live paediatric imaging, is far less established in the post-mortem setting. There are only a few articles either describing single site experience,[10] or focussing on single anatomical areas such as congenital cardiac disorders.[11] With rising costs, radiographer staffing shortages and reduced availability and access to MRI and CT imaging, ultrasound may play a key role in a future non-invasive post-mortem imaging service. The equipment is cheaper to acquire and run than cross-sectional imaging, with a variety of healthcare specialists already skilled in sonographic techniques (eg, fetal medicine clinicians, obstetricians, radiologists, sonographers). Nevertheless, without robust evidence detailing the diagnostic accuracy of this technique, it remains difficult to counsel parents appropriately and for clinicians to understand the benefit of this modality as a non-invasive imaging alternative to conventional autopsy. The aim of this systematic review is

therefore to comprehensively assess the diagnostic accuracy of paediatric post-mortem ultrasonography (PMUS) from the existing literature.

## MATERIALS AND METHODS

This study was registered in PROSPERO International prospective register of systematic reviews, CRD42018106968.[12] Preferred Reporting Items for Systematic reviews and Meta-Analyses (PRISMA) guidelines for transparent reporting of systematic reviews were followed.[13] Ethical approval was not required for this retrospective review of published data.

### Patient involvement

Patients were not directly involved in the design of this study.

### Literature review

We searched MEDLINE (Ovid), EMBASE and the Cochrane Library databases for eligible articles published between 1 January 1998 and 31 December 2018 (20-year range), using database specific Boolean search strategies with terms and word variations relating to 'autopsy', 'ultrasonography' and 'paediatrics'. The full search strategy was conducted in December 2018 (See online supplementary appendix S1 for details). A repeat search was conducted in July 2019 did not reveal any further eligible manuscripts for inclusion.

### Eligibility criteria

Inclusion criteria encompassed work investigating diagnostic accuracy of post-mortem ultrasound (PMUS) imaging using autopsy as a reference standard. Studies were limited to human subjects, including fetuses (any gestation) and children (aged 0–18 years). No restrictions were placed on method of ultrasound technique, machine vendor and experience of operator or type of clinical setting. No language restrictions were used.

Exclusion criteria included studies reported as conference abstracts, case reports, editorials, opinion articles, pictorial reviews, multimedia files (online videos, podcasts) and small case series (fewer than five cases). Articles without autopsy reference standard, or those relating to diagnostic accuracy for antenatal ultrasound imaging were excluded.

All articles were independently searched by two reviewers (both paediatric radiologists with>8 years (SCS) and >12 years (OJA) experience). Abstracts of suitable studies were examined, and full papers were obtained. References from the retrieved full-text articles were manually examined for other possible publications. Disagreements were resolved by consensus.

### Data extraction and quantitative data synthesis

Two reviewers (SCS and OJA) independently extracted data from the full articles into a database (Excel, Microsoft, Redmond WA, USA) which included the following factors: study design, study setting, population demographics, sample size, index test, study methodology, use of reference standard, diagnostic accuracy outcomes and measure of bias.

Raw numbers of true and false positives and negative PMUS diagnoses were collected and inputted into a 2×2 table, considering autopsy as reference standard. Missing data were calculated if possible and, unpublished data or further information was obtained by contacting the authors.

### Methodological quality

The quality and measure of bias for each included study was assessed by the modified Quality Assessment of Diagnostic Accuracy Studies (QUADAS-2) criteria.[14] Disagreements were resolved by consensus review of the literature.

### Statistical analysis

Statistical analysis was performed by an experienced statistician (DL) using R.[15] The available data were used to compute sensitivity and specificity for each study, with a 0.5 correction factor for studies with zero frequencies. A bivariate random-effects model was used to obtain combined mean estimates of sensitivity and specificity with 95% CIs for overall PMUS diagnosis and, by three body systems (neurological, cardiothoracic and abdominal). The functions *madad* and *reitsma* within the R package *mada* was used to synthesise the evidence. This bivariate approach takes into account the correlation between sensitivity and specificity when different threshold values are used for diagnosis in each study/ body system.[16] Summary receiver operating characteristic (SROC) curves were presented to show the relationship between the two outcomes. Spearman's correlation coefficient (rho) was calculated to test for the presence of different thresholds.

## RESULTS

### Eligible studies

The initial search performed on 1 January 2019 yielded 6017 articles, after removal of duplicate studies. On the basis of study title and abstract, 5999 articles were excluded and 18 articles were retrieved in full text. After review of the full text, four studies were eventually included in the systematic review.[17–20] A PRISMA flowchart is shown in figure 1.

### Patient population

Our patient and study characteristics are provided in tables 1 and 2. All four articles related to PMUS for perinatal deaths.[17–20] In all studies, parental consent was given for the ultrasound imaging. In total, 495 fetuses were imaged (sample size range: 75–169) and of these, 455 underwent an autopsy as a reference standard. The gestational ages ranged from 11 to 48 weeks. Most of the fetuses were the result of terminations of pregnancy (363/495, 73%), with 11% (52/495) miscarriages and

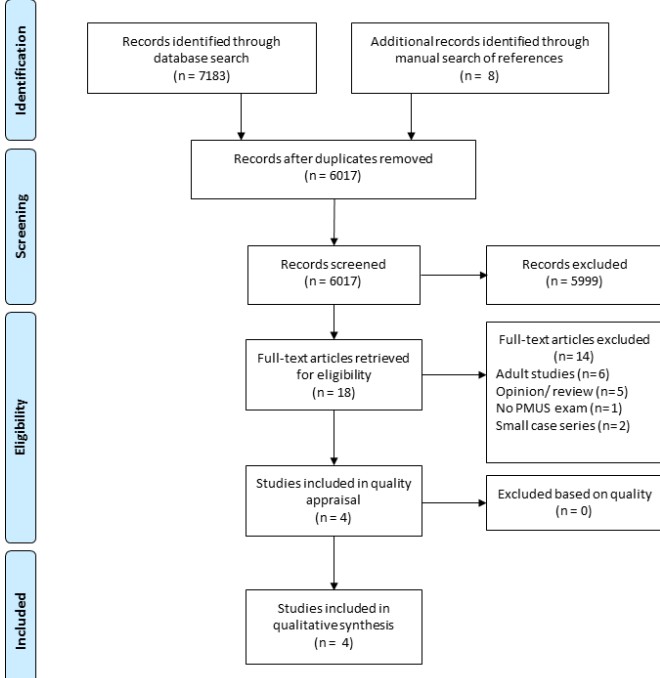

**Figure 1** Preferred Reporting Items for Systematic reviews and Meta-Analyses flow chart for the study search and selection.

16% (80/495) intrauterine deaths. Gender mix was not reported in any of the studies.

The majority of studies (3/4, 75%) were prospective in design and conducted in a single centre (3/4, 75%). Two studies originated from France,[18 19] one study originated from Belgium[20] and one was a multicentre study involving patients from two institutions, one in the UK and one in Belgium.[17] The timeframe for data collection ranged between 1 and 4 years.

### Imaging assessment and operator

Details regarding the index (PMUS) and reference (autopsy) tests in the studies of this systematic review are summarised in tables 3 and 4.

A variety of ultrasound machine vendors and models were used. The most popular ultrasound machine was Voluson E8 (GE Medical Systems, Zipf, Austria), used by 3/5 (60%) centres. High-frequency linear probes were used by all operators, with 3/5 (60%) centres also using curvilinear or micro-convex ultrasound probes to acquire two-dimensional (2D) images. One study also acquired three-dimensional (3D) ultrasound volumes for head and body imaging in addition to standard 2D views.[20]

A single operator performed all PMUS studies in almost all centres, except for one centre (based in Belgium) for the study by Kang *et al*,[17] where multiple operators performed the ultrasound scanning. For all cases, the ultrasound was performed prior to autopsy (therefore without knowledge of the reference test results). In two studies (2/4, 50%), the operators were blinded to the antenatal history,[17 19] and in the remainder the antenatal results were known to the operator.[18 20] The operator was

**Table 1** Demographic details of population studied for all articles

| Author, year | Country | Sample size, n | Patient group | Median gestation at death | Gestational age range (weeks) | Median post-mortem weight (g) | Weight range (g) | Mode of death |
|---|---|---|---|---|---|---|---|---|
| Prodhomme et al, 2015[10] | France | 169 | Fetuses | 27 weeks | 15–38 | Not stated | Not stated | 97% (164) terminations 3% (5) intrauterine deaths |
| Tuchtan et al, 2018[19] | France | 75 | Fetuses | Not stated, however 42 cases were<24 weeks; 33 were>24 weeks | 15–38 | Not stated | Not stated | 79% (59) terminations 4% (3) miscarriages 17% (13) intrauterine deaths |
| Votino et al, 2018[20] | Belgium | 88 | Fetuses | 21 weeks | 11–40 | 702 | 7–4020 | 66% (58) terminations 17% (15) miscarriages 17% (15) intrauterine deaths |
| Kang et al, 2019[17] | Belgium and UK | 163 imaged, 123 with reference test | Fetuses | 23 weeks | 13–42 | Not stated | Not stated | 50% (82) terminations 21% (34) miscarriages 29% (47) intrauterine deaths |

**Table 2** Study characteristics for articles included in systematic review

| Author, year | Country | Sample size | Study design | Number of centres | Patient selection | Study period | Index test | Reference test |
|---|---|---|---|---|---|---|---|---|
| Prodhomme et al, 2015[10] | France | 169 | Retrospective | Single | Unclear | 4 years (2009–2013) | 2D whole-body ultrasound | Conventional autopsy |
| Tuchtan et al, 2018[19] | France | 75 | Prospective | Single | Consecutive | 1 year (2014) | 2D whole-body ultrasound | Conventional autopsy |
| Votino et al, 2018[20] | Belgium | 88 | Prospective | Single | Consecutive | 19 months (2012–2013) | 2D and 3D whole-body ultrasound | Conventional autopsy |
| Kang et al, 2019[17] | Belgium and UK | 163 | Prospective | Multiple, two centres | Consecutive | 2 years (2014–2016) | 2D whole-body ultrasound | Conventional and minimally invasive autopsy |

2D, two-dimensional; 3D, three-dimensional.

frequently a paediatric radiologist (3/5, 60%) or fetal medicine specialist with ultrasound experience (2/5, 40%).

### Reference test
For all studies, the reference standard was conventional autopsy. Only one study described a subset of patients (11/123, 9%) who underwent a minimally invasive autopsy by endoscopic examination,[17] using a technique described by Sebire et al.[21] All of the studies reported that their autopsies were performed according to national guidelines (ie, Royal College of Pathologists,[22 23] SOFFoeT,[24] French National Authority of Health).[25] In one study, the autopsy guideline was not mentioned.[18]

All autopsies were performed by one of several pathologists within a department, rather than a single individual. The pathologists were aware of the clinical and antenatal history for all cases, and in three of the four studies the pathologists were blinded to the ultrasound results.

### Methodological quality assessment
Using the QUADAS-2 tool, there was a moderate to high risk of bias concerning the 'index text' domain because the persons performing the ultrasound examination in two studies were aware of the patient's antenatal history, and therefore could have been biassed when performing the examination.[18 20] The risk of bias regarding the 'flow and timing' domain were unclear for two other studies since timing between ultrasound examination and autopsy were not defined.[18 19]

In one study,[18] the risk of bias was unclear for both the patient selection and also reference standard, given that it was not stated how patients were recruited into the study or whether the autopsy results could have been biassed by the ultrasound findings. Assessment of bias regarding overall applicability concerns was low (figure 2).

### Diagnostic accuracy of ultrasound by body system
The diagnostic accuracy rates for sensitivity and specificity of each body system are summarised in tables 5 and 6, with the results given in SROC plots in figures 3 and 4 for overall body diagnoses, and per body system, respectively.

Two studies did not explicitly state the non-diagnostic imaging or non-diagnostic autopsies rate.[18 19] Kang et al[17] reported a non-diagnostic ultrasound result for 13/70 (18.6%) brain, 21/123 (17.1%) thoracic, 24/122 (19.7%) cardiac and 19/123 (15.4%) abdominal examinations. The non-diagnostic autopsy rate was 53/123 (43.1%) for the brain and 1/123 (0.8%) for the heart.

Votino et al[20] reported non-diagnostic ultrasound imaging in 4/62 (6.5%) brain and 2/86 (2.3%) cardiothoracic ultrasound examinations. The non-diagnostic autopsy rate was 17/88 (19.3%) for the brain. In 5/88 (5.7%) cases, the brain was not examined at autopsy, only the body.

Four studies were included in the bivariate meta-analysis model, all related to ultrasound usage in

**Table 3** Details of index test for studies included for systematic review

| Author, year | Ultrasound machine | Ultrasound transducers/probes | Imaging time (min) | Patient preparation | Ultrasound operator, experience | No. operators | Blinded to clinical history | Time between delivery to imaging | Diagnostic accuracy subgroup measures |
|---|---|---|---|---|---|---|---|---|---|
| Prodhomme et al, 2015[10] | Philips iU22 | 5–8 MHz microconvex, 5–12 MHz linear, 5–17 MHz linear | Not stated | No additional preparation over cold storage of body | Paediatric radiologist, >10 years of experience | Single | No | Not stated | Agreement with final autopsy diagnosis. List of diagnoses given |
| Tuchtan et al, 2018[19] | Toshiba Aplio 500; Supersonic Aixplorer; GE Voluson E8 | Toshiba: 7–10 MHz curved, 10–15 MHz linear; Supersonic: 6–10 MHz curved, 11–15 MHz linear; GE: 5–10 MHz curved, 10–14 MHz linear | 20 | No additional preparation over cold storage of body | Paediatric Radiologist, 15 years ultrasound and 8 years post-mortem imaging experience | Single | Yes | Not stated | Anatomical structures were divided into seven categories: brain, spine, lung, heart, skeletal, gastrointestinal and genitourinary. Overall and individual body organ accuracy rates given |
| Votino et al, 2018[20] | GE Voluson E8 | 6–18 MHz linear, 6–12 MHz curved, 5–9 MHz curved | 15 | Fetuses either fixed in formalin (>15 weeks gestation), otherwise no additional preparation. | Fetal medicine doctor, 10 years of ultrasound experience | Single | No | Median time 2 days (1 hour–4 days) | Diagnoses categorised into three body systems: neurological (brain/spine), thorax (including heart) and abdomen. Individual body system accuracy rates given |
| Kang et al, 2019[17] | GE Voluson E8; GE LOGIQ E9; Samsung HM70A | GE Voluson: 6–18 MHz linear, 6–12 MHz curved, 5–9 MHz curved; GE Logiq: 2.5–8 MHz linear, 1–5 MHz curved; Samsung: 7–16 MHz linear | Not stated | No additional preparation over cold storage of body | Fetal medicine doctor or paediatric radiologist (>5 years experience each) | Two (one from each centre) | Yes | Median time 2 days (0–39 days) | 19 internal organs assessed and grouped into four anatomic regions for analysis: neurological (brain/spine), thorax, heart and abdomen. Overall and individual body system accuracy rates given |

**Table 4** Details of the reference test for studies included for systematic review

| Author, year | Reference test | Autopsy standards | Person(s) performing reference test, experience | Blinded to antenatal clinical history | Blinded to index (ultrasound) test results | Median time between imaging or delivery and autopsy |
|---|---|---|---|---|---|---|
| Prodhomme et al, 2015[10] | Conventional autopsy | Not stated | Pathologists, experience not quantified | No | Not stated | Not stated |
| Tuchtan et al, 2018[19] | Conventional autopsy | Societé française de foetopathologie (SOFFoeT, France) guidelines | Pathologists, experience not quantified | No | Yes | Not stated |
| Votino et al, 2018[20] | Conventional autopsy | Societé française de foetopathologie (SOFFoeT, France) guidelines | Pathologists with 20 years of experience | No | Yes | 1 day from US to autopsy (1 hour–2 days) |
| Kang et al, 2019[17] | Conventional autopsy or minimally invasive autopsy | Conventional autopsy guidelines (SOFFoeT and Royal College of Pathologists) Or Minimally invasive endoscopic methods described by Sebire N et al, 2012 | Paediatric pathologists, >15 years of experience | No | Yes | 5 days from delivery to autopsy (0–47 days) |

perinatal deaths. There was little evidence that studies had different thresholds for diagnosis (ie, negative correlation between sensitivity and specificity) (Spearman's correlation (rho)=−0.310; 95% CI: −0.670 to 0.168). The mean diagnostic sensitivity and specificity, respectively (reported in table 6) was 84.3% (95% CI: 70.8% to 92.2%) and 96.7% (95% CI: 86.5% to 99.3%) for neurological abnormalities; 52.1% (95% CI: 27.6%

to 75.5%) and 96.6% (95% CI: 86.8% to 99.2%) for cardiothoracic abnormalities and 78.4% (95% CI: 61.0% to 89.4%) and 97.3% (95% CI: 88.9% to 99.4%) for abdominal abnormalities. Combining all body systems, the mean sensitivity was 73.3% (95% CI: 59.9% to 83.5%) and the mean specificity was 96.6% (95% CI: 92.6% to 98.4%). Mean diagnostic sensitivity in the cardiothoracic body system was significantly lower than the neurological system (p=0.010) and marginally lower than the abdominal system, although non-significant at the 5% level (p=0.059).

The study by Kang et al[14]14 appeared to be an outlier compared with the other three studies, particularly for specificity rates of all body systems. When excluding this outlier the mean diagnostic sensitivity and specificity respectively was 87.4% (95% CI: 79.5% to 92.5%) and 97.9% (95% CI: 90.7% to 99.6%) for neurological abnormalities; 57.8% (95% CI: 24.1% to 85.6%) and 98.7% (95% CI: 94.5 to 99.7) for cardiothoracic abnormalities and 82.3% (95% CI: 63.8% to 92.5%) and 98.4% (95% CI: 94.5% to 99.5%) for abdominal abnormalities. Combining all body systems, the mean sensitivity was 78.8% (95% CI: 64.6% to 88.4%) and the mean specificity was 98.3% (95% CI: 96.3% to 99.2%). See online supplementary figures S2 and S3 for the SROC curves for overall diagnoses and by body system, with this study excluded.

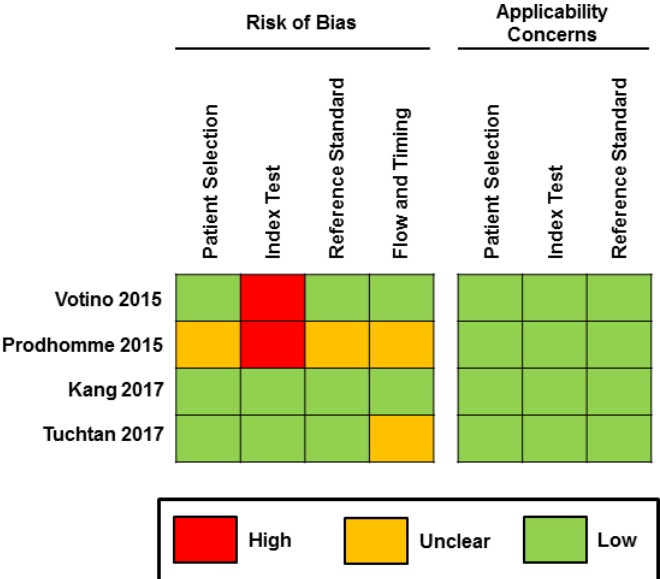

**Figure 2** Methodological quality assessment of the included studies using the Quality Assessment of Diagnostic Accuracy Studies 2 tool. Risk of bias and applicability concerns summary about each domain are shown for each included study.

## DISCUSSION

PMUS for perinatal deaths is a feasible technique, and currently performed in a few European centres. The

**Table 5** Estimated sensitivities and specificities of different body parts/systems from studies included in systematic review, with 95% CI in square brackets

| Author, year | Body part | Total sample size | Sample size with autopsy | Excluded ND autopsy | Excluded ND PMUS | Excluded No autopsy | TP | FN | FP | TN | Sensitivity, % (95% CI)* | Specificity, % (95% CI)* |
|---|---|---|---|---|---|---|---|---|---|---|---|---|
| **Neurological** | | | | | | | | | | | | |
| Votino et al, 2018[20] | Brain | 88 | 66 | 17 | 4 | 5 | 10 | 1 | 7 | 48 | 87.5 (59.8 to 97.1) | 86.6 (75.3 to 93.2) |
| Prodhomme et al, 2015[10] | Brain | 169 | 169 | 0 | 0 | NA | 84 | 10 | 0 | 75 | 88.9 (81.1 to 93.8) | 99.3 (94.0 to 99.9) |
| Tuchtan et al, 2018[19] | Brain | 75 | 75 | 0 | 0 | NA | 22 | 5 | 1 | 47 | 80.4 (62.4 to 91.0) | 96.9 (87.8 to 99.3) |
| Prodhomme et al, 2015[10] | Spine | 169 | 169 | 0 | 0 | NA | 18 | 0 | 0 | 151 | 97.4 (79.1 to 99.7) | 99.6 (96.9 to 100.0) |
| Tuchtan et al, 2018[19] | Spine | 75 | 75 | 0 | 0 | NA | 10 | 0 | 0 | 65 | 95.5 (67.9 to 99.5) | 99.2 (93.1 to 99.9) |
| Kang et al, 2019[17] | Brain and spine | 123 | 70 | 53 | 13 | NA | 13 | 8 | 11 | 38 | 61.4 (40.8 to 78.5) | 77.0 (63.7 to 86.5) |
| **Cardiothoracic** | | | | | | | | | | | | |
| Prodhomme et al, 2015[10] | Thoracic | 169 | 169 | 0 | 0 | NA | 10 | 1 | 0 | 158 | 87.5 (59.8 to 97.1) | 99.7 (97.1 to 100.0) |
| Kang et al, 2019[17] | Thoracic | 123 | 123 | 0 | 21 | NA | 5 | 14 | 20 | 84 | 27.5 (12.8 to 49.4) | 80.4 (71.9 to 87.0) |
| Tuchtan et al, 2018[19] | Thoracic | 75 | 75 | 0 | 0 | NA | 6 | 2 | 0 | 67 | 72.2 (40.2 to 91.0) | 99.3 (93.3 to 99.9) |
| Prodhomme et al, 2015[10] | Cardiac | 169 | 169 | 0 | 0 | NA | 2 | 9 | 0 | 158 | 20.8 (6.7 to 49.1) | 99.7 (97.1 to 100.0) |
| Kang et al, 2019[17] | Cardiac | 123 | 122 | 1 | 24 | NA | 13 | 13 | 18 | 78 | 50.0 (32.4 to 67.6) | 81.0 (72.0 to 87.5) |
| Tuchtan et al, 2018[19] | Cardiac | 75 | 75 | 0 | 0 | NA | 4 | 18 | 0 | 53 | 19.6 (8.3 to 39.5) | 99.1 (91.7 to 99.9) |
| Votino et al, 2018[20] | Cardiac and thoracic | 88 | 88 | 0 | 2 | NA | 16 | 2 | 5 | 65 | 86.8 (65.5 to 95.8) | 92.3 (83.7 to 96.5) |
| **Abdominal** | | | | | | | | | | | | |
| Votino et al, 2018[20] | Whole abdomen | 88 | 88 | 0 | 2 | NA | 12 | 2 | 4 | 70 | 83.3 (58.4 to 94.7) | 94.0 (86.2 to 97.5) |
| Kang et al, 2019[17] | Whole abdomen | 123 | 123 | 0 | 19 | NA | 17 | 11 | 23 | 72 | 60.3 (42.4 to 75.9) | 75.5 (66.0 to 83.0) |
| Prodhomme et al, 2015[10] | Gastrointestinal | 169 | 169 | 0 | 0 | NA | 15 | 10 | 0 | 144 | 59.6(40.7 to 76.0) | 99.7 (96.8 to 100.0) |
| Tuchtan et al, 2018[19] | Gastrointestinal | 75 | 75 | 0 | 0 | NA | 2 | 0 | 1 | 72 | 83.3 (31.0 to 98.2) | 98.0 (91.7 to 99.5) |
| Prodhomme et al, 2015[10] | Urinary system | 169 | 169 | 0 | 0 | NA | 37 | 2 | 0 | 130 | 93.7 (81.8 to 98.0) | 99.6 (96.4 to 100.0) |
| Tuchtan et al, 2018[19] | Urinary system | 75 | 75 | 0 | 0 | NA | 9 | 1 | 0 | 65 | 86.4 (57.1 to 96.8) | 99.2 (93.1 to 99.9) |

*Sensitivities, specificities and associated 95% CI are calculated using the standard univariate approach while making an adjustment of 0.5 for zero frequencies where necessary.
FN, false negative; FP, false positive; NA, not applicable; ND, non-diagnostic; PMUS, post-mortem ultrasound; TN, true negative; TP, true positive.

**Table 6** Estimates of mean sensitivity and specificity (with 95% CIs), overall and split by different body systems

| Body system | Mean sensitivity (95% CI) | Mean specificity (95% CI) |
|---|---|---|
| Overall | 73.3 (59.9 to 83.5) | 96.6 (92.6 to 98.4) |
| Neurological | 84.3 (70.8 to 92.2) | 96.7 (86.5 to 99.3) |
| Cardiothoracic | 52.1 (27.6 to 75.5) | 96.6 (86.8 to 99.2) |
| Abdominal | 78.4 (61.0 to 89.4) | 97.3 (88.9 to 99.4) |
| P value (neuro vs cardio) | 0.010 | 0.916 |
| P value (neuro vs abdom) | 0.498 | 0.819 |
| p-value (cardio vs abdom) | 0.059 | 0.731 |

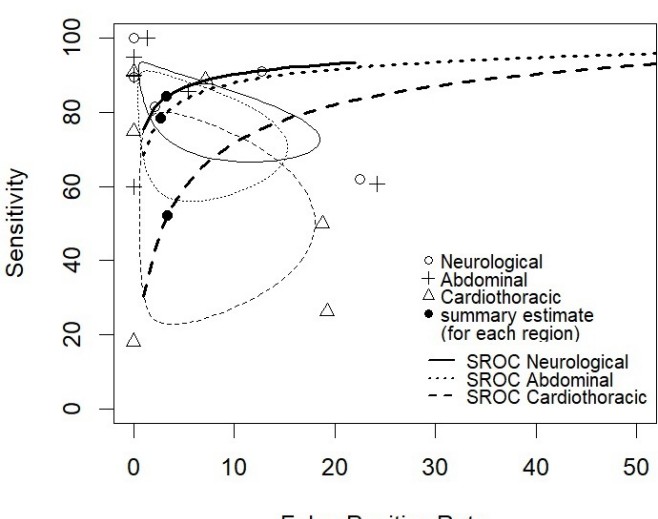

**Figure 4** Receiver operating characteristic plot of sensitivity against false positivity rate (1, specificity) of 19 studies from table 5 separated by body system. Bivariate overall summary estimates of sensitivity and false positivity rate for each body system are overlaid with corresponding 95% confidence ellipses. SROC, summary receiver operating characteristic

diagnostic accuracy rates are highest for neurological and abdominal abnormalities (mean sensitivity rates of 84.3% and 78.4%, respectively), but less effective for cardiothoracic abnormalities (mean sensitivity rate of 52.1%). There were no studies reporting the PMUS accuracy rates in neonates and older children.

The study raises two main potential clinical implications, first relating to improved parental choice and second potential healthcare cost savings. It is well known that parents report a high acceptability for non-invasive autopsy methods,[3 26 27] and that many healthcare professionals also advocate post-mortem imaging techniques.[28] Having another imaging modality available, such as ultrasound, would likely increase options for parental choice and their access to imaging. It may serve as a means for opening channels of communication between parents and clinicians, possibly reduce time spent grieving by

offering a diagnosis and form of closure, or highlight the need for further counselling. As pregnant mothers will ubiquitously have been subject to fetal or antenatal ultrasound, they may have better inherent understanding of PMUS rather than cross-sectional imaging techniques.

In terms of finances, a previous systematic review by Ahmad et al[4] described a potential cost saving with MRI virtopsy compared with invasive autopsy of approximately 33% (based on a full body MRI costing £226.34 and an autopsy costing £471.80, with a paediatric MRI virtopsy sensitivity rate of 73%). Using their same reference for costing,[29] an ultrasound study taking approximately 20 min (costing £56 per case, with similar whole-body ultrasound pooled sensitivity rate of 74%) could offer even further benefits. Nevertheless, this calculation is based on assuming a similar rate of referral for invasive autopsy, which may be unlikely given the potential higher parental acceptability. Whether the balance between increased referrals and reduction in cost per patient would negate potential savings remains to be seen.

The comparatively low PMUS sensitivity rates for cardiothoracic abnormalities is also important to recognise and may warrant alternative imaging with higher sensitivity (e,g, PMMR; 60%–82%).[30 31] Whether detection of abnormalities with ultrasound will improve with operator experience, or is related to the technique itself is unknown. The lack of circulation and inability to use Doppler imaging on ultrasound, teamed with the presence of post-mortem intracardiac gas may hamper sonographic visualisation of cardiac structural abnormalities but 3D PMUS may possibly resolve some of these issues in future studies. In addition, PMUS operators may undergo a learning curve similar to that found for thoracic abnormalities using PMMR, with improvements in diagnostic

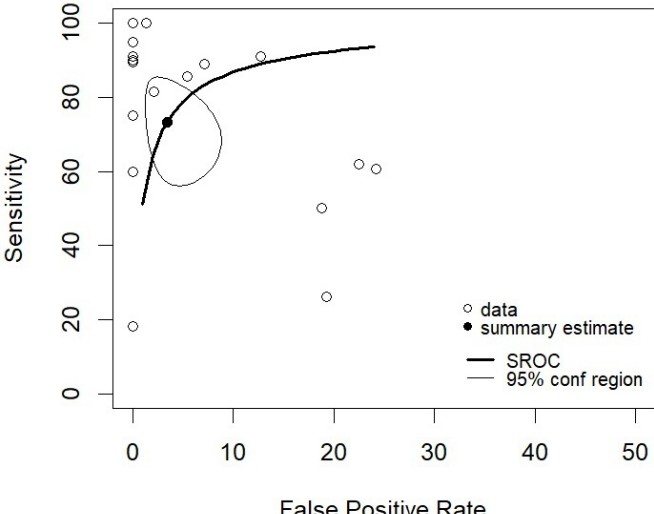

**Figure 3** Receiver operating characteristic plot of sensitivity against false positivity rate (1, specificity) of all 19 studies from table 5. Bivariate overall summary estimates of sensitivity and false positivity rate are overlaid with corresponding 95% confidence ellipses. SROC, Summary receiver operating characteristic.

accuracy from 30%–37% in early reports[32] to 81.8%[31] later. This may relate to better knowledge of normal post-mortem changes on imaging (ie, not overcalling pleural effusions as pathological) although infective pathologies still remain difficult to detect.

In this systematic review, the majority of studies reported similar ultrasound diagnostic accuracy rates although those by Kang et al[17] were much lower, particularly for sensitivity rates of neurological and thoracic abnormalities and, specificity rates for all body systems. Reasons for this may relate to the fact that non-diagnostic ultrasound results were considered false negative for sensitivity calculation, and false positives for specificity calculations. In addition, the authors did report a high non-diagnostic ultrasound and autopsy rate for brain imaging of approximately 13/70 (18.6%) and for thoracic imaging of 21/123 (17.1%). While demographic details of the perinatal deaths in their study were similar to others, the high non-diagnostic rate may suggest a greater referral of macerated fetuses, which could have impeded ultrasound visualisation of certain organs and thus accurate diagnosis. There was also a laparoscopic-assisted minimally invasive autopsy as the reference standard in a subset of cases (11/123, 9%).[21] While this technique has been shown to provide a high success rate in terms of tissue sampling and detection of histological abnormalities, the true diagnostic accuracy rate compared with conventional autopsy remains unknown.[33] Despite these factors, we still include the results in our meta-analysis given that this was the only multicentre trial in our review and described results from more than one ultrasound operator, thereby providing potentially a more generalisable result for PMUS outcomes.

There were several limitations to this study. The first relates to the generalisability of our results. There were very few studies assessing the diagnostic accuracy of PMUS in children, with all studies focussing on perinatal deaths, conducted within Europe, and with ultrasound examinations being performed by the same operator within each centre. The examinations were conducted within a few specialist centres focussing on this niche area, meaning they may represent an elevated level of accuracy, potentially not achievable if PMUS was to be offered in a more general clinical setting.

The second limitation includes the classification and detection of abnormalities. In all cases, the pathologists performing the autopsies were not blinded to the antenatal history, and in one of the studies the ultrasound operator was also unblinded. There may have been a variation in the way PMUSs are performed and reported. For example, some operators may have considered ascites and pleural effusions as part of normal post-mortem change,[34 35] while others may view these as significant abnormalities. There may also have been variation regarding the recording of non-diagnostic ultrasound examinations. In two studies, the number of non-diagnostic autopsies and ultrasound examinations were not reported,[18 19] and it is not clear whether those

ultrasound examinations were all of diagnostic quality, whether these were excluded prior to analysis, or whether the authors simply recorded these as 'no abnormality' where no significant findings were found, regardless of diagnostic quality. In the other two papers where non-diagnostic ultrasound examinations were reported, one excluded these from analysis,[17] whereas the other considered them as an inaccurate finding (ie, recorded as 'false positive').[20] Nevertheless despite such limitations, our work has several strengths. This paper represents the first systematic review on use of PMUS in perinatal deaths. It demonstrates a reasonable diagnostic accuracy for the majority of body systems and that it can be performed on a variety of scanner models, in a varied cohort of perinatal deaths across a range of gestational ages.

Currently, there are no standardised protocols on how best to perform a PMUS study, nor for which situations in would be most beneficial in helping to change clinical management for future pregnancies. For example, it may have an apparent higher diagnostic rate in fetal demise for those pregnancies who have not yet undergone detailed second or third trimester antenatal ultrasound (ie, after 20 weeks gestation).

Further research should be direct towards PMUS studies involving a larger cohort of perinatal deaths and also assessing usage in older children. There is clearly potential for this technique to be adopted in different clinical settings (such as in the developing world where pathological expertise is lacking but where causes of death are still important to confirm to inform global public health initiatives,[36] by other specialists to increase appeal and availability following adequate training (eg, pathologists, obstetricians, sonographers) and international guidelines should be established for the appropriate indications for usage. There is also the ability for ultrasound to be used as a tool to aid percutaneous organ biopsies where tissue sampling is required to confirm histological diagnoses. In conclusion, this systematic review has provided evidence in showing that it is a feasible technique with a reasonable diagnostic accuracy when performed by experienced operators; however, further work will enable a better understanding of its place and purpose within a future non-invasive autopsy service.

**Contributors** All authors listed in this manuscript fulfil the ICMJE recommendations for authorship. All authors provided substantial contribution to the conception and design of the work, analysis and interpretation; and drafting the work for intellectual content. All authors have had final approval for the version to be published and agree to be accountable for all aspects of the work in ensuring that questions related to the accuracy are appropriately investigated and resolved.

**Funding** SCS is supported by a RCUK/ UKRI Innovation Fellowship and Medical Research Council (MRC) Clinical Research Training Fellowship (Grant Ref: MR/R00218/1). This award is jointly funded by the Royal College of Radiologists (RCR). OJA is funded by a National Institute for Health Research (NIHR) Career Development Fellowship (NIHR-CDF-2017-10-037), and NJS funded by an NIHR Senior Investigator award. The authors receive funding from the Great Ormond Street Children's Charity and the Great Ormond Street Hospital NIHR Biomedical Research Centre. This article presents independent research funded by the MRC, RCR, NIHR and the views expressed are those of the author(s) and not necessarily those of the NHS, MRC, RCR, the NIHR or the Department of Health.

**Competing interests** None declared.

**Patient consent for publication** Not required.

**Ethics approval** Institutional review board approval was not required because it comprises a systematic review of published literature.

**Provenance and peer review** Not commissioned; externally peer reviewed.

**Data availability statement** Data are available upon reasonable request. All data relevant to the study are included in the article or uploaded as supplementary information.

**ORCID iD**
Susan Shelmerdine http://orcid.org/0000-0001-6642-9967

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
