## [Reviewer comments · BMJ Paediatrics Open]

ARTICLE DETAILS

TITLE (PROVISIONAL)	Diagnostic Accuracy of Perinatal Post-mortem Ultrasound (PMUS): A Systematic Review
AUTHORS	Shelmerdine, Susan; Langan, Dean; Sebire, Neil; Arthurs, Owen

VERSION 1 – REVIEW

REVIEWER	Reviewer name: Sarah Nevitt Institution and Country: University of Liverpool United Kingdom Competing interests: I have no competing interests
REVIEW RETURNED	23-Aug-2019

GENERAL COMMENTS	I have performed a statistical review of the manuscript "Diagnostic Accuracy of Paediatric Post-mortem Ultrasound (PMUS): A Systematic Review" The authors present a systematic review and meta-analysis of four studies measuring the diagnostic test accuracy of PMUS compared to autopsy as the reference standard. Overall the paper is very well written in terms of the clarity of the terminology used, both in terms of the clinical context and the methodology. However, there is an important error within the methodology employed here, which impacts upon the numerical results and therefore the conclusions of the meta-analysis. The authors pool sensitivity and specificity values within the four studies separately using a meta-analysis method for pooling proportions. While sensitivity and specificity certainly can be considered as proportions as a data type, the problem within meta-analysis of these values is that they are mathematically related. I.e. within the 2x2 table often constructed for a DTA study, any change in the values which will result in a change to sensitivity has to result in a change to specificity too if the total number of patients is kept constant. Therefore, pooling these values independently may lead to underestimation of test accuracy (see Deeks 2011 for further details). For this reason, specific meta-analysis techniques have been developed which take account of the mathematical relationship of sensitivity and specificity and allow values of pooled sensitivity and pooled specificity to be estimated. These methods do not allow a measure of statistical heterogeneity such as I-squared to be used, but it is generally accepted by design that DTA studies are heterogeneous so methods tend to use random-effects to adjust for this as standard.
---

	I suggest that the most appropriate method for this particular question is the Bivariate model described by Retisma et al 2005 (see ref below). This model can be implemented using PROC NLMIXED in SAS statistical software (please see Chapter 10 of the Cochrane Handbook for Diagnostic Test Accuracy Studies for example SAS code). I would be happy to review the results and conclusions of this manuscript again if the authors are able to repeat the analysis using a meta-analytic method suitable for DTA studies (such as the method I have suggested above). I also have a couple of minor wording comments: Page 4, line 53: the word 'without' has a 3 in it Page 5: It is good that the authors have used PRISMA guidelines for this work. I suggest that the authors should specifically be following the extension for DTA studies (see reference) and for transparency, including a PRISMA DTA checklist with their resubmission Results: Do I understand correctly that a 'true positive' in this context would be the same abnormality being detected by both PMUS and autopsy? Or any abnormality detected by both? Similarly, would a 'true negative' be that neither PMUS nor autopsy detect any abnormality? References Deeks JJ. Systematic reviews in health care: Systematic reviews of evaluations of diagnostic and screening tests. BMJ 2001; 323: 157-162. McInnes MDF, Moher D, Thombbs BD et al. Preferred Reporting Items for a Systematic Review and Meta-analysis of Diagnostic Test Accuracy Studies: The PRISMA-DTA Statement. JAMA. 2018;319(4):388-396 Reitsma JB, Glas AS, Rutjes AW, Scholten RJ, Bossuyt PM, Zwinderman AH. Bivariate analysis of sensitivity and specificity produces informative summary measures in diagnostic reviews. J Clin Epidemiol 2005; 58: 982-990.
--	--

REVIEWER	Reviewer name: TUCHTAN Lucile Institution and Country: Assistance Publique Hôpitaux de Marseille Hôpital de la Timone France Competing interests: no
REVIEW RETURNED	26-Aug-2019

GENERAL COMMENTS	thank you for this interesting review on use of post mortem ultrasound in perinatal deaths. just few comments: p 4: ligne 53: without not "wi3thout" table 3 p12 ligne 35: single table 4 p 14: Tuchtan use the guidelines SOFFoet too
---

REVIEWER	Reviewer name: Xin Kang Institution and Country: University Hospital Brugmann, Brussels, Belgium Competing interests: Fetal post-mortem imaging (MRI, PMUS)
REVIEW RETURNED	02-Sep-2019

GENERAL COMMENTS	Congratulation with your manuscript " Diagnostic accuracy of paediatric post-mortem ultrasound (PMUS): a systematic review"! This is the first systematic review concerning the use of high-frequency ultrasound for perinatal post-mortem examination. The paper is well written and the review is conducted using a clear methodology. - I have noted one typing error: page 4 line 53 "without" - Additionally, concerning the paper from Kang et al, PMUS was performed by multiple operators in one center and by one operator in another center. The lower sensitivity and specificity is explained by the fact that non-diagnostic results were considered false negative for sensitivity calculation, and false positive for specificity calculation (cf higher sensitivity and specificity reported by the same group in a paper published in 2019 comparing PMUS and PMMRI, in which only diagnostic cases were analyzed). This information is written currently in the limitations of the study. Maybe this point could be clarified in the paragraph discussing about the different results obtained by Kang et al (page 21 line 16-41). - Finally, considering that the 4 reviewed papers concerned perinatal PMUS, I suggest to add "peri-natal" in the title.
--

VERSION 1 – AUTHOR RESPONSE

Reviewer 1: Sarah Nevitt:

1. There is an important error within the methodology employed here, which impacts upon the numerical results and therefore the conclusions of the meta-analysis. I suggest that the most appropriate method for this particular question is the Bivariate model described by Retisma et al 2005. This model can be implemented using PROC NL MIXED in SAS statistical software (please see Chapter 10 of the Cochrane Handbook for Diagnostic Test Accuracy Studies for example SAS code. I would be happy to review the results and conclusions of this manuscript again if the authors are able to repeat the analysis using a meta-analytic method suitable for DTA studies

Thank you for your recommendation. We initially considered the bivariate analysis approach, but wrongly assumed correlation could not be present since there is no explicit threshold value to be chosen in these diagnoses. However, it is possible some readers are more 'trigger happy' in making diagnoses than others and therefore the 'threshold' may still be present.

Our statistician (DL) is not a SAS user, and therefore applied the same methods you recommend using the R package mada. You will find the bivariate analysis results in the updated manuscript.

2. Page 4, line 53: the word 'without' has a 3 in it

Thank you – this has now been amended.

3. Page 5: It is good that the authors have used PRISMA guidelines for this work. I suggest that the authors should specifically be following the extension for DTA studies (see reference) and for transparency, including a PRISMA DTA checklist with their resubmission

Thank you – we have now included a PRISMA DTA checklist for both the article and also for the abstract (as per the PRISMA statement website). This has been uploaded in the editorial manager as supplementary material.

4. Results: Do I understand correctly that a ‘true positive’ in this context would be the same abnormality being detected by both PMUS and autopsy? Or any abnormality detected by both? Similarly, would a ‘true negative’ be that neither PMUS nor autopsy detect any abnormality?

Yes, you understand correctly that a true positive is the same finding detected by both PMUS and autopsy, not any abnormality. Similarly a true negative means that neither PMUS nor autopsy detected any abnormal findings.

Reviewer 2: Lucile Tuchtan

5. Page 4: line 53: without not "wi3thout"

6. Table 3 p12 ligne 35: single

Thank you for spotting the typographical errors in points 5 and 6 above – these have all been corrected.

7. Table 4 p 14: Tuchtan use the guidelines SOFFoet too

We now include that Tuchtan used the SOFFoet guidelines in their autopsy criteria, now in Table 4.

Reviewer 3: Xin Kang

8. I have noted one typing error: page 4 line 53 "without"

Thank you – this has now been amended.

9. Additionally, concerning the paper from Kang et al, PMUS was performed by multiple operators in one center and by one operator in another center. The lower sensitivity and specificity is explained by the fact that non-diagnostic results were considered false negative for sensitivity calculation, and false positive for specificity calculation (cf higher sensitivity and specificity reported by the same group in a paper published in 2019 comparing PMUS and PMMRI, in which only diagnostic cases were analyzed). This information is written currently in the limitations of the study. Maybe this point could be clarified in the paragraph discussing about the different results obtained by Kang et al (page 21 line 16-41).

Thank you for this excellent point – we have now included the point regarding non-diagnostic studies being included as ‘false negatives/positives’ into our discussion and mention the other point regarding different operators in our results section of the systematic review.

10. Finally, considering that the 4 reviewed papers concerned peri-natal PMUS, I suggest to add "peri-natal" in the title.

Thank you – our title has now been amended.

VERSION 2 – REVIEW

REVIEWER	Reviewer name: Sarah Nevitt Institution and Country: University of Liverpool, United Kingdom Competing interests: I have no competing interests
REVIEW RETURNED	08-Oct-2019
GENERAL COMMENTS	Thank you to the authors for their responses and their efforts in addressing my statistical comments. I am happy that the revised methodological approach using a bivariate random effects model and employed within R using the madad and reitsma functions is appropriate. I also consider that the presentation and interpretation of results is appropriate. I am happy to recommend this manuscript for publication